# A Dynamical Systems Approach to Characterizing Brain–Body Interactions during Movement: Challenges, Interpretations, and Recommendations

**DOI:** 10.3390/s23146296

**Published:** 2023-07-11

**Authors:** Derek C. Monroe, Nathaniel T. Berry, Peter C. Fino, Christopher K. Rhea

**Affiliations:** 1Department of Kinesiology, University of North Carolina at Greensboro, Greensboro, NC 27402, USA; 2Under Armour, Inc., Innovation, Baltimore, MD 21230, USA; 3Department of Health and Kinesiology, University of Utah, Salt Lake City, UT 84112, USA; 4College of Health Sciences, Old Dominion University, Norfolk, VA 23508, USA

**Keywords:** dynamical systems, sample entropy, recurrence quantification analysis, electroencephalography, neuromotor control, oscillatory activity, biophysical models

## Abstract

Brain–body interactions (BBIs) have been the focus of intense scrutiny since the inception of the scientific method, playing a foundational role in the earliest debates over the philosophy of science. Contemporary investigations of BBIs to elucidate the neural principles of motor control have benefited from advances in neuroimaging, device engineering, and signal processing. However, these studies generally suffer from two major limitations. First, they rely on interpretations of ‘brain’ activity that are behavioral in nature, rather than neuroanatomical or biophysical. Second, they employ methodological approaches that are inconsistent with a dynamical systems approach to neuromotor control. These limitations represent a fundamental challenge to the use of BBIs for answering basic and applied research questions in neuroimaging and neurorehabilitation. Thus, this review is written as a tutorial to address both limitations for those interested in studying BBIs through a dynamical systems lens. First, we outline current best practices for acquiring, interpreting, and cleaning scalp-measured electroencephalography (EEG) acquired during whole-body movement. Second, we discuss historical and current theories for modeling EEG and kinematic data as dynamical systems. Third, we provide worked examples from both canonical model systems and from empirical EEG and kinematic data collected from two subjects during an overground walking task.

## 1. Introduction

Dynamic, coordinated movements require complex signaling between the central nervous system and effectors (i.e., skeletal muscles, joints, and limbs). These brain–body interactions (BBIs) are studied using tools at the interface of engineering and neuroscience to address impactful questions that cut across disciplines, from uncovering the origins of consciousness to the development of brain–computer interfaces for rehabilitation and human performance. In the most general sense, studies of BBIs fall into one of three categories based on their treatment of each variable: (i) studies manipulating movement to characterize changes in the brain; (ii) studies manipulating brain circuits to characterize changes in movement; (iii) studies directly characterizing the nature of BBIs by quantifying relationships between brain and body signals. The goal of this review is to serve as a primer for scientists interested in the third perspective, quantifying BBI as temporal dependencies between systems. First, we explicate on the methodological and theoretical challenges associated with measuring and interpreting scalp-measured neural oscillatory activity (using electroencephalography) during dynamic movement. Second, we motivate and showcase two solutions for modeling interactions between ‘brain’ and ‘body’ systems that are yet unexplored in this growing field.

## 2. EEG Signal Acquisition and Preprocessing

A primary consideration for jointly studying BBIs must be the means by which signals from these systems are acquired. For example, a major challenge for studies of BBIs is that movement itself is a serious confound for practically all brain imaging techniques, including functional magnetic resonance imaging (fMRI) and electroencephalography (EEG), the two techniques that are most often used to study BBIs. Compared to fMRI, devices that collect scalp-measured EEG readings are more cost-effective, carry a significantly smaller footprint, and sample data at a much higher resolution (>250 Hz), all factors contributing to their widespread use in studies of BBIs. Although the use of EEG readings is increasingly ubiquitous, there are few standards concerning the pre-processing or ‘cleaning’ of EEG data collected during dynamic movements. While the possibilities for processing scalp-measured EEG readings are practically infinite, they can be broadly classified based on the ultimate analytic goal. On one hand, the signal can be analyzed as a series of time-locked positive and negative deviations around events of interest (e.g., a heel strike or movement initiation), which are commonly referred to as ‘event related potentials’ (ERPs). Gwin and colleagues [1] published some of the earliest recommendations for improving the veracity of ERPs acquired during treadmill walking and running. They demonstrated that regressing out artifactual components from EEG data acquired using a high-density (248-channel) array revealed ERPs during running that were identical to ERPs in response to a visual oddball discrimination task acquired while standing at rest. Indeed, this general approach is still widely used, even though BBI studies are increasingly using limited montages (64 channels or fewer), which limits the number of artifacts that can be identified. BBI studies have also increasingly taken a different analytic approach, treating EEG data as a continuous amalgam of oscillatory activity representing dynamic brain circuit interactions rather than discrete, event-locked perturbations. New analytic tools for this task are published often, which can be overwhelming to novice investigators and non-experts. In the following sections, we outline two of the most frequently employed pre-processing steps (re-referencing and artifact removal) and highlight existing recommendations for their use in studies that use scalp-measured EEG readings to study BBIs.

### 2.1. Re-Referencing

The poor spatial resolution of EEG signals can be attributed in large part to volume conduction—the attenuation of current flow by heterogeneous tissues between the source and recording electrode. This also contributes to the small amplitude of scalp EEG signals (~100–200 μV), since the signals are spread from their source indiscriminately across the entire scalp and are thus ‘canceled’ by use of the common reference. It is possible to improve two-dimensional spatial resolution—or confidence that sensor-level signals are generated by proximal sources—by performing a re-referencing step, typically offline during preprocessing. One common method is to perform a surface Laplacian (i.e., current scalp density) transform, which is essentially a high-pass spatial filter that is generally considered to be reference- and assumption-free [2]. By filtering out signal content from distal sources, superficial, cortical sources are isolated (in μv/mm^2^). At the same time, the resulting current density maps are preferential to dipoles oriented perpendicular to the scalp, from the gyri, and less sensitive to dipoles oriented tangential to the scalp, from the sulci, which may be detrimental to the study of movement [3]. Importantly, for the filter to perform optimally, electrodes should be evenly spaced over the scalp. This is common for research-grade EEG devices, but it is not standard for consumer-grade devices, such as those often used in BBI studies, which typically contain fewer electrodes arranged unevenly over smaller portions of the scalp (e.g., the forehead or over the top of the head).

A popular alternative to the surface Laplacian is to set all channels relative to a common average reference. This is based on the theory that common potentials sampled over a volume containing all current sources will be fully canceled out [4]. A benefit is that the units of the signal are not transformed and remain intuitive (μV). However, there is some evidence that the limited montages most commonly used in the study of BBI (i.e., those with ≤32 electrodes) are not sufficiently dense to accurately sample the spatial distribution of ‘whole-volume’ potentials, thus violating the basic assumption underlying this approach [5]. If data cannot be re-referenced, then it is challenging to make inferences about the underlying brain circuits that give rise to those signals with any spatial certainty, and grand averaging over all electrodes may be preferred. Following suit with a recent set of recommendations for so-called ‘mobile brain–body imaging’ [6]—which at the time of this publication remains on a preprint server and has been cited by three different published empirical studies—we would advise that when EEG data are recorded using ≥32 channels that are evenly distributed (e.g., in a 10–10 or 10–20 configuration), investigators should tend toward average referencing their data. If fewer channels are used, then the surface Laplacian can be used. If fewer channels in a non-uniform montage are used, then grand averaging to obtain a single EEG timeseries is warranted.

### 2.2. Artifact Removal

Although some EEG experts recommend minimal signal processing for studies of ERPs [7], there is strong evidence that EEG signals are universally contaminated by non-brain artifacts, including electrical activity from proximal muscles and mechanical interference from electrode and cable movement [8,9]. Removing and imputing particularly noisy EEG channels is a widely accepted practice for improving global signal-to-noise. Although it is typically easy to identify and remove bad channels manually due to their small amplitudes (~100–200 uV), artifacts induced during dynamic movement make this task more challenging. One alternative is to determine channel quality by the correlation between a channel and its robust estimate based on other channels, and this approach is built into the *clean_rawdata* function implemented in the popular EEGLAB toolbox. Although a threshold for the correlation between a channel and its prediction of r < 0.8 is the default, this is likely too high if a high proportion of channels contain noise. In experiments with high degrees of contamination, this method loses its utility, as the channel prediction is itself inaccurate. The use of dry, active electrodes—a popular tool for BBI experiments—for which impedances cannot be quantified and therefore signal quality cannot be ensured during acquisition represents another possible use case where this approach may not perform well. Furthermore, if limited montages (e.g., <32 electrodes) are being used, then removing a few channels may significantly reduce scalp coverage.

On the other hand, blind source separation methods allow for the removal of artifactual components of a signal while potentially preserving channels. Independent component analysis (ICA) is the most commonly used method for statistically disentangling mixed components of neural and non-neural origin. One recent implementation, adaptive mixture independent component analysis (AMICA), has proven particularly well-suited for EEG signals collected during dynamic movements, outperforming a more traditional implementation (i.e., InfoMax) [10]. Regardless of the algorithm, the number of components returned by an ICA are restricted by the rank of the data, meaning that EEG signals recorded from more (clean) channels provide for greater degrees of freedom. Notwithstanding, AMICA appears to perform sufficiently with 35 channels, but suffers when the number of channels is reduced to 25 [11]. Once components are identified, they can be removed through manual classification by expert reviewers, or subject to automated algorithms such as ICLabel [12], which recovers components comparably between mobile and stationary data collection, particularly for montages containing 64 or more channels [13].

Artifact subspace reconstruction (ASR), a PCA-based method for artifact removal [14] (also built into the *clean_rawdata* function in EEGLAB), is a different flavor of blind source separation that is gaining popularity. Although ASR seemingly struggles to remove regularly occurring artifacts, such as eye blinks, it may complement ICA-based approaches that are better suited for that task [15,16]. However, ASR requires a ‘clean’ reference period to be recorded and there is no clear agreement in the proper method for selecting cut-off values.

Ultimately, both a channel-wise removal step and a blind-source separation step are practically requirements for BBI-EEG studies. If speed is a factor or if the EEG data have been acquired using ≤25 channels, then the InfoMax algorithm is probably sufficient. Otherwise, AMICA is well-suited for ‘noise’ structures that are inherent in EEG signals collected during dynamic movement. Further, to maintain appropriate rank of the data, channel interpolation (of removed channels) should only be performed after the second blind-source separation step. The use of ASR in conjunction with an ICA-based algorithm is warranted, but users should note that it is common for the algorithm to identify windows (‘bursts’) that cannot be corrected. In epoch-based analyses, it is possible to trim or ignore these, but such non-continuous data streams pose a particular problem for the study of dynamical systems (as defined in this manuscript); therefore, ASR should be employed judiciously depending on the ultimate goal of the analytic pipeline.

## 3. Interpretation of EEG-Measured Oscillatory Activity

It is generally accepted that scalp-measured EEG rhythms represent dynamic post-synaptic membrane potentials that create dipoles, which are spatially distinct areas of positive charge (i.e., in the cell body) and negative charge (i.e., in the dendrites) [17]. Scalp-measured oscillatory activity is typically quantified and described based on oscillatory activity in different canonical frequency bands: slow (1–7 Hz), alpha (8–14 Hz), beta (15–30 Hz), and low gamma (30–50 Hz) rhythms. Since the first spontaneous EEG recording by Hans Berger in the 1920′s, clinical and scientific pursuits have been dominated by attempts to relate cortical rhythms in these bands to complex behaviors. These bands are not specific to motor or sensory processes, meaning that their interpretation relies on the circuits from which they emerge: (i) orientation of those circuits with respect to the scalp and other circuits, (ii) proximity with respect to the scalp and other circuits, and (iii) (a) synchronous firing of neuronal populations that is, in turn, coordinated by complex intracortical and thalamic projections [18]. When viewed through this lens, the ‘generators’ of these rhythms are not particular cells or areas (e.g., anterior cingulate cortex, thalamus), but changes in (i) membrane properties of neurons, (ii) synaptic processes between neurons, (iii) dynamic short- and long-range connectivity between discrete elements of neuronal networks, (iv) neuromodulation of any of these features by neurotransmitter systems [19].

These basic tenets led to the widely cited concept of event-related (de)synchronization (ERD/ERS), which states that decreases and increases in spectral power after onset of a stimulus are due to the desynchronization and synchronization, respectively, of some or all these mechanisms [20]. However, there are two issues that preclude the use of this framework in the study of BBI. First, a ‘baseline’ or ‘stimulus-free’ state is necessary for proper interpretation of ERD/ERS, yet this state is difficult to define if the goal is to model BBIs. Second, and highly related to the themes of this review, it is increasingly understood (i) that the pre-stimulus state of neuronal networks influences (or moderates) their response to tasks and (ii) that the event-related interactions between these circuits are neither purely linear nor purely periodic [21,22]. For the most part, in BBI studies using scalp-measured EEG data, a priori feature selection is based on prior findings from the BBI literature (e.g., a focus on slow-rhythm oscillations during walking without consideration for faster rhythms) and on the response of those features to cognitive demands (e.g., alpha rhythms during an attentional task). Instead of considering the psychological correlates of scalp-measured neural oscillatory activity, we take the perspective that these rhythms contain information that can be interpreted in terms of the circuits (and circuit interactions) that generated them. This biologically grounded perspective is likely superior if the ultimate goal of understanding BBIs is to develop better diagnostic or prognostic tools or inform more effective interventions for movement and psychiatric disorders alike. In the following sections, we aim to distill data from biophysical models that elucidate more precisely the circuit interactions that give rise to these rhythms.

### 3.1. Brain Dynamics: From Spiking to Scalp-Measured Oscillations

A core challenge for interpreting EEG signals, given the movement- and non-movement-related sources of interference described above, is interpreting the neurobiological mechanisms of scalp-measured, neuroelectric oscillations. Historically, neuroscientists believed that neuron spiking behaviors, representing action potentials of that single neuron, were the base code for communication in the brain [23]. It is now widely accepted that (de)synchronization of those neurons, as can be recorded from oscillations in the local field potential (LFP; recorded from populations of neurons), are inherently important to understanding dynamic brain states and thus communication [24,25]. These oscillations contain rich information about synaptic and dendritic signaling mechanisms [26]. LFPs are not merely epiphenomena, and the global propagation of these oscillations across the brain appear to alter distant local activity in meaningful ways [27].

While there is long-standing evidence that surface EEG monitoring (i.e., multichannel arrays embedded directly in the cortex) may contain LFP-like information [28], the same biophysical properties of the cortex, dura, cerebrospinal fluid, and skull that give rise to volume conduction (and the poor spatial resolution of scalp-measured EEG) also challenge the notion that scalp-measured EEG data contain anything LFP-like. An EEG electrode on the scalp does not record spiking or even LFPs directly, but instead captures differences in electrical potential caused by current flow induced by many synchronized post-synaptic potentials [29]. This, of course, is not a mechanism; it explains the existence of the scalp-measured oscillatory activity in which we are ultimately interested; however, the relationship between LFPs and scalp-measured EEG signals remains poorly understood [30]. Ground-breaking work is being performed in this space by relating primate EEG and microelectrode recordings (i.e., to capture single-unit, multi-unit, and LFP signals). One recent novel study reported patterns of visual processing in macaque EEG signals that contained similar information to concomitant microelectrode recordings and also looked similar to human MEG signals during a similar task [31]. However, whether spiking or LFP activity recorded from primates during movement can be teased out from concomitant scalp-based EEG data remains unknown.

Indeed, much of the cited literature that follows—to decompose the circuit interactions giving rise to scalp-measured oscillatory activity by frequency band—does not include dynamic movement, with most of the movement being restricted to finger tapping or similar. It is possible that this is inconsequential, as movement intentions are coded and programmed in similar ways (e.g., population coding), in similar places (e.g., primary and premotor cortex), and maintained/terminated by similar circuitry (e.g., basal ganglia, cerebellum), at least in terms of the spatial resolution of scalp-measured EEG signals. On the other hand, this may also be viewed as a critical limitation of the literature and therefore, a call-to-action for scientists to tease out the differences between BBIs during tasks with different motor demands. If true, then the summary of literature that follows should serve as an important benchmark to begin testing whether and to what degree the interactions giving rise to these rhythms are different during dynamic, full-body movements, such as walking, and the tools introduced in the second half of this tutorial review may be useful for these efforts.

### 3.2. Delta/Theta (1–7 Hz)

The EEG spectrogram is dominated by oscillatory activity in these low frequencies, perhaps unsurprising given that they are generally conceptualized as cortex-specific rhythms originating from interactions between cortical (layer V) pyramidal neurons [32]. Thus, these slow cortical oscillations plausibly orchestrate long-range cortical interactions, entraining and modulating the amplitude of faster rhythms that regulate short-range, local circuits [33,34]. Because these frequencies are substantially slower than neuronal firing rates, they likely originate from synaptic interactions and represent cellular processes such as short-term depression/potentiation and spike-time-dependent plasticity, making them of great interest to those studying learning and memory [35]. Slow cortical oscillations occur on a timescale that more closely approximates cognition, such as attention and motor planning, and have been most commonly observed in transmodal brain regions—particularly the prefrontal cortex—that are understood to be especially important for these processes [36,37].

Rodent studies of ‘active sensing’ suggest that during ambulation and exploration of novel environments, these rhythms represent selective tuning of cortical receptive fields that, in turn, integrate sensory information and govern movement planning [38]. Supporting this conceptualization, Sipp and colleagues [39] used high-density EEG signals and source localization to reveal greater slow rhythm (4–7 Hz) spectral power in dorsolateral-prefrontal and anterior cingulate regions during balance beam walking compared to treadmill walking, with even greater increases observed during periods of instability on the balance beam. These findings represent some of the strongest support for the interpretation of scalp-measured slow rhythms as representing sensorimotor integration processes, such as error detection. However, interpreting these rhythms as representing something ‘brain’-specific may not be so straightforward. It has been suggested that EEG signals within 15 harmonics of the fundamental stepping frequency (~1 Hz) are contaminated during dynamic movement, meaning that the preprocessing recommendations summarized above are particularly salient for analyses seeking to quantify BBIs using slow cortical rhythms [8]. Notwithstanding, one study compared intracranial brain signals from six epileptic patients during different phases of a ‘center-out’ motor task—a paradigm that obviates the stepping frequency issue—and demonstrated pronounced increases in slow rhythm (2–4 Hz) spectral power during the movement planning phase, but not during movement itself [40].

### 3.3. Alpha (8–14 Hz)

Alpha rhythms were the first to be identified by visual inspection of EEG signals and remain the most well-studied rhythms. One of the most prominent features in the awake EEG is the increased amplitude of alpha power over the occipital lobe (visual cortex) during eyes-closed resting conditions, which led to the widespread notion of a posterior alpha ‘generator’ and the theory that these rhythms represent brain idling [41]. However, both concepts are increasingly contested, as alpha rhythms have been observed over practically the entire scalp [42], sometimes even increasing in response to task demands [43]. This encourages a viewpoint introduced by Grey Walter, an early EEG pioneer, that scalp-measured alpha rhythms represent a spatial average of an impossibly large number of circuit interactions (‘generators’) [44] and thus, the strength of alpha in an EEG recording is dependent on the synchronous activity of many generators over a large distance. In the context of BBIs and the challenges associated with volume conduction, this lends toward a ‘global’ approach to alpha band spectral power.

On the other hand, in considering a more local source, oscillatory activity in this range that is measured over the central sulcus (‘rolandic area’) is commonly specified as a ‘mu’ rhythm. When measured directly from the cortex itself (using electrocorticography) mu appears to originate in extragranular cortical layers (Layers I–III) and represent a regional flow of information, from higher-order associative regions to the sensorimotor cortex, that collectively represent cortico-thalamocortical feedback loops [45]. Thus, changes in mu rhythms can be interpreted to represent net changes in activity along these loops, either in the whole brain volume or in local circuits, depending on the re-referencing approach used. Likewise, in the context of complex and dynamic movement, it has been proposed that mu rhythms may serve as a biomarker of predictive coding: a process of matching motor planning schemas against sensory and proprioceptive information [46,47]. Reductions in mu power with increasing gait speed and a concomitant shift toward slow cortical oscillations may represent greater cortical involvement and a priming of the sensorimotor cortex to receive and respond to sensory feedback, supporting this conceptualization of alpha/mu [48]. Supporting patterns have been observed across the gait cycle, with mu power increasing as the contralateral foot pushes off the ground and decreasing during the contralateral foot’s swing phase [48,49].

### 3.4. Beta (15–30 Hz)

Changes in beta rhythms during movement are widely reported in the mobile imaging literature and beta rhythms have consistently been localized to the sensorimotor cortex [50,51]. Alpha/mu and beta are both suppressed during volitional movement [52,53] and enhanced after movement termination [54]. These patterns have supported an interpretation of beta as a biomarker of cortical ‘idling’, similar to alpha/mu [41]. A broader interpretation is that beta rhythms represent delayed information transmission across cortical circuits and thus, movement suppression [55]. One of the most rigorous biophysical models of transient cortical beta rhythms to date suggests they arise from ‘bottom-up’ thalamo-cortical projections in a proximal pathway carrying sensory information to granular/infragranular layers (layers IV and V) [56]. The authors speculate that beta-mediated delays in transmission could be due to either enhanced inhibition or excessive excitation in supragranual layers, preventing top-down information flow (i.e., in alpha/mu rhythms). That same model suggests an additional distal pathway that dampens excitability of a broader cortical area via synapses in upper, supragranular layers. Thus, it is possible to conclude that beta rhythms in layers I–III are driven by thalamic bursts which would in turn suppress top-down information flow as represented by reduced alpha/mu. This conceptualization of a ‘beta lock’ on information flow is supported by the faster recovery of beta rhythms and slower recovery of alpha rhythms after movement termination [54].

### 3.5. Low Gamma (30–50 Hz)

The power-law scaling of the EEG power spectra means that gamma rhythms (30–90 Hz) exhibit the smallest amplitudes, making them somewhat difficult to discern in scalp-based EEG readings. EEG signals are also contaminated by ‘power line’ noise at 50 or 60 Hz due to the cycle rate of alternating current powering all the devices and lighting in the vicinity of measurement. Therefore, it is common to restrict analyses to this lower end of the gamma range. Gamma rhythms have been a target of intense scrutiny because of their alignment with spiking patterns from local cell ensembles [57,58] that are, in turn, governed by interactions between inhibitory (GABA) and excitatory (Glutamatergic) synapses. Although the balance between these inhibitory and excitatory neurons is important, it is indeed the GABA-ergic input to these circuits which are most important for maintaining a balanced state of excitability or criticality [59]. Scalp-measured gamma rhythms are believed to arise from either (i) a clock-like mechanism in inhibitory-inhibitory circuits or (ii) reciprocally connected inhibitory-excitatory circuits [58]. These circuits exist uniformly over the entire cortex; thus, methods for resolving the EEG signal in space, either through re-referencing or source localization, are critical for the meaningful interpretation of BBI using these rhythms. A few studies have reported gait-related modulation of low gamma [60,61,62,63], even though it is commonly observed that EMG-measured muscle activity demonstrates oscillatory activity in the same range, suggesting that the scalp-measured changes in low-gamma during dynamic movement may be artifactual.

## 4. A Dynamical Systems Approach to Brain–Body Interactions

Studies of BBI to date have relied on statistical dependencies between brain and body signals [64]. While these efforts have revealed novel and interesting patterns of (a)synchrony, they generally lack grounding in established theories of motor control. For example, the well-recognized generalized motor program theory states that motor control is a function of motor programs that are embedded within the central nervous system [65] and variability in task execution is considered skill/performance error. In contrast, dynamical systems theory states that motor control is regulated by nonlinear interactions between biological subsystems [66]. From a mathematical sense, dynamical systems theory examines how systems evolve over time, defining the state of the system by some set of variables, and the behavior of that system by a set of rules. Thus, in the context of human locomotion, dynamical systems theory implies that motor output is dependent on internal and external conditions and variability in the system represents stability/instability in the system. Traditional correlational approaches to quantifying BBIs are insufficient for capturing these complexities.

The differences between these approaches are not merely mathematical; they represent divergent ways of thinking about motor behavior. For example, in the classical sense, movement is believed to arise from a pattern generator or collection of controllers. This leads to an interpretation of movement variability (e.g., stride time variability during over ground walking, reaction time variability in response to a cognitive task) as representing error arising from maladaptive organization of the underlying cognitive or anatomical structures (controllers). On the other hand, a dynamical systems approach leads to an interpretation that some degree of variability is ‘normal’, representing adaptive system organization [67,68]. The differences become even more nontrivial when considering how to relate measures of each system (e.g., EEG-measured oscillatory activity and accelerometer-measured movement kinematics). Movement patterns are not governed strictly by output from the nervous system, but by information, which is, in turn, generated from a recurrent network engaging sensory and motor anatomical networks and cognitive processes. The interactions between systems are inherently nonlinear and are characterized by metastability, representing flexibility of the system around an attractor state, which cannot be captured using the traditional linear (‘effect = structure’) approach [69]. This acknowledgement led to growth in the fractal physiology field [70,71], which adopts a dynamical systems approach to examine changes within and between systems. Therefore, the focus and foundation of thought presented herein is grounded in dynamical systems theory.

If the goal is to understand control between the brain and body, then a multitude of time- and response-scales must be considered, with an eye toward methodological considerations to match theory, measurement, and interpretation. The behavior of dynamical systems can be described by the motion towards/around an attractor, or, a location within the state space that a system evolves over time. More specifically, a point attractor is a value that describes a single point of stability, a limit cycle is a set of points that a system is attracted to within the state space, and a strange attractor is an attractor with sensitivity to initial conditions [72,73,74]. Dynamical systems are deterministic and chaotic if their behavior is sensitive to initial conditions. A dynamical system with noise becomes a stochastic process, meaning that the system itself follows a random probability distribution. Previously, dynamical systems theory has described and characterized biological systems, including the brain [75,76] and body [68,77,78]. More specifically, limit cycles and strange attractors have been used to model and describe human locomotion [76,78]. For a more comprehensive overview of dynamical systems, readers are directed elsewhere [72,73,74].

Within the study of BBI, there may be specific and identifiable linkages that define the coupling between any two (or more) systems. Within the context of dynamical systems theory, coupled oscillators refer to two (or more) oscillators that work in a manner in which they can transfer energy/information between them. Synchronization of these coupled oscillators occurs when these two systems become interlocked in frequency or phase [79]. Oscillations are found throughout biological systems [80] and the synchronization of oscillators has also been examined [81].

It is worth noting that the goal to model interpretable linkages between systems may seem obvious to some readers but contrasts against the goal of studies attempting to develop models merely for predictive purposes, as is common in the brain–computer interface (BCI) literature. For example, one seminal BCI study reported that artifact-free event-related oscillatory components (akin to combining the ‘event-related’ and ‘continuous’ approaches introduced earlier) during a walking task could serve as inputs into a recurrent neural network that would accurately recreate movement kinematics [82]. Assuming that the components were indeed artifact-free, this is an intriguing finding that (as the authors note) aligns with what is known about neuromotor control, given that the components were roughly centered over the motor cortex. However, these approaches do not necessarily give insight into the nature of the interactions between systems because of the inherent ‘black-box’ nature of connections in deep learning networks.

### 4.1. Modeling Brain–Body Interactions

A *model* is any representation of reality and can be presented in visual/symbolic, qualitative, or quantitative forms. While each of these are applicable to BBI, we will focus on quantitative models. These quantitative models can be further subdivided into a variety of categories, including modeling of the data and modeling the system. Modeling the data provides a method of characterizing the dynamics of BBI, with the objective of providing a concise description of the underlying patterns within two systems. Modeling the system refers to an analytical approach that aims to characterize and examine the dynamics of a system, or systems, based on a priori knowledge. These models may be static or dynamic, deterministic or stochastic, time-invariant or time-variant, linear or nonlinear, continuous or discrete, among others. There is little doubt that these models have a place within the study of BBI, but our focus within this review will remain strictly around modeling the data to provide a quantitative assessment of the system(s).

In addition to the necessary processing steps required for each independent signal, several additional considerations must be made. Throughout the study of brain–body interactions, a mixing of deterministic and stochastic signals is inevitable. For instance, EEG data contain a fractal structure with power-law scaling [83,84,85] and limit cycles have been used to characterize joint angles throughout varying tasks [86,87]. Within the field of biomechanics, and specifically those interested in gait, measure of stride time, width, and length have also been used to describe the dynamics of human locomotion [88,89,90]. These measures, similar to that of EEG, are not deterministic but do contain fractal structure.

The state space reconstruction of any single signal that contains complex and nonlinear structure can be performed by choosing the appropriate time delay and embedding dimension. The embedding dimension, *m*, is the number of embeddings to retain and the time delay, *τ*, is the distance in time that each of these embeddings should be separated. However, any two systems, or the measures being taken from any two systems, may function on two completely scales of response. From a modeling perspective, these asymmetries pose specific challenges, however, these nonlinearities are the focus of interest. If the response scale between two measures of interest is asymmetric, rescaling these data is essential to subsequent analyses.

### 4.2. Dynamics of a Single System

The state space reconstruction of a single system is a foundational component of many analytical techniques used to describe the behaviors of dynamical systems. Recreation of the state space with a single time series can be performed using Takens’ theorem [91]. This process requires a determination of the time delay, *τ*, and embedding dimension, *m*. The time delay is often chosen using either the autocorrelation function (ACF) or average mutual information (AMI) of the time-delayed time series. The ACF has the advantage of a simpler calculation; however, AMI does not rely on linear correlations, which makes it the recommended choice for chaotic systems [92,93]. Regardless of the method chosen, the time delay should be close enough in time that it loosely approximates a derivative but far enough in time that they are not repetitive [94].

Calculation of the embedding dimension, *m*, is a nontrivial task that can significantly impact the state space reconstruction of the system. The false nearest neighbors algorithm [95] and Cao’s algorithm [96] are two methods of estimating the embedding dimension. Within the subsequent sections, we utilize the false nearest neighbors algorithm to inform the choice of *m*. However, we acknowledge that the false nearest neighbors algorithm is impacted by both noise [95,97] and data length [98]. With implementation on stochastic signals, the calculation of false nearest neighbors will often not reach 0%, requiring interpretation by the researcher. With *τ* and *m*, the state space reconstruction through Takens’ theorem can be performed and visualized.

Surrogate data are often used to confirm the nonlinear and dynamic structure of the original time series [99]. Surrogate processes include a variety of techniques, including random shuffle, Gaussian number generation, and phase randomization.

Approximate entropy provides a measure of system complexity, including both deterministic and stochastic processes [100]. Sample entropy is similar to the approximate entropy statistic but offers a few important advantages [101]. Firstly, the sample entropy statistic addresses an issue of self-matching inherent to the approximate entropy calculation. Secondly, the sample entropy statistic is less sensitive to data length [102]. Similar to approximate entropy, a lower value indicates more self-similarity, or a more deterministic signal, compared to a time series with a higher value.

Recurrence plots [103] and recurrence quantification analysis [104] provide a means of reducing higher dimensional attractors to two dimensions, analogous to the way that graph theory can be used to reduce multidimensional network interactions to more manageable scalar values. Recurrence quantification plots can provide visualizations of the dynamics of the system, including aspects of nonstationarity as well as periodicities and determinism within a signal. Further, these plots, and the quantitative analysis of these plots, provide robust analysis of deterministic and stochastic signals. The generation of a recurrence plot requires the reconstruction of the attractor (i.e., trajectory matrix) through time delay methods (i.e., autocorrelation function or average mutual information). A tolerance, *r*, value is also required to determine a specified minimum number of neighbors, enhancing the visualization of the system and the quantification of the plot space.

### 4.3. Coupling between Two Systems

Quantifying the coupling, or interdependence, between systems has an obvious level of importance across many areas of medicine and the biological sciences. Examining the topological mappings of two systems within the reconstructed state space can provide a good visual of interdependence, represented by continued and close proximity between the two systems in time and space. Although we will not cover these concepts in depth, mutual or cross-predictions made from one system to the other, visualized within the state space, can indicate interdependence between the systems.

Extensions of the approximate entropy, sample entropy, and recurrence quantification analyses that include crossed time series also exist. Similar to the univariate case, cross-approximate entropy and cross-sample entropy are regularity statistics that provide indices of coupling between two time series [100,101,105]. When reconstructing the state space of a single system, it is necessary to determine the optimal parameters of *L* and *m*, however it becomes necessary to determine mutually agreeable values for these parameters when examining coupled systems. In this case, the cross autocorrelation function and mutual false nearest neighbors algorithms are commonly used.

Because of the nature of both cross-ApEn and cross-SampEn from an *m* and *r* perspective, it is essential that both are calculated on the standardized time series. Standardization of the time series, in the simplest case, can refer to subtracting the mean and dividing by the standard deviation. However, other transforms may be applied to one or both of the time series.

Cross recurrence analysis (Cross RQA) is an extension of the univariate case, allowing for the examination of the dynamic relations between two signals [104,106]. One advantage of cross RQA, particularly in example cases such as the examination of brain–body interactions, is that it does not require equal sample lengths. Thus, in cases where data are collected at different sampling frequencies, cross recurrence analysis provides a means of examining the dynamics of the two systems before any resampling techniques might be performed to create time-matched data.

## 5. Worked Examples

To make the concepts discussed above more concrete and approachable, we provide examples of coupling between (i) ‘known’ systems of predetermined dynamics and (ii) empirical data collected from EEG and inertial measurement units (IMUs) during an overground walking task.

### 5.1. Known Systems: Van Der Pol Oscillators

This section is separated into two key parts. The first section includes a progression of simulated data, beginning with a simple harmonic oscillator, building to the use of a limit cycle with nonlinear damping, and concluding with a mathematical example of the synchronization of two coupled systems.

Within the first example (Figure 1), we utilize sine and cosine functions with varying amplitudes and imposed drift. The state space reconstructions of these systems are provided in the leftmost column, with the system responses across time provided in the middle column, and the recurrence plots in the rightmost column. As a baseline, two sine waves are shown concurrently within the top row of figures, followed by an adjustment to the amplitude of one of the signals in the second row. An adjustment to the amplitude of one of the functions results in an obvious difference in the state space reconstruction, which subsequently removes any recurrences from the cross-recurrence plot (using the same radius). Similarly, shifting the mean upwards of the amplitude-adjusted system (i.e., third row) similarly impacts the cross-recurrence quantification. These two examples highlight the importance of considering the scale of the systems being considered. Further, drift in a signal was simulated in the fourth row. This drift (fourth row, middle column) provides a similar response to issues related to data acquisition or baseline changes in the responses of a system. The result, or consequences, of this drift on the cross-recurrence plot further highlights the importance of the acquisition and processing steps.

Building on the first example, the second example utilizes the Van der Pol system. The Van der Pol system has been commonly used to model biological systems such as cardiac dynamics but has also been used within the gait literature to simulate stride intervals [107,108] and neuronal firing patterns [109]. The second set of examples (Figure 2) include the state space reconstructions of the systems in the leftmost columns with the system response across time in the middle column and cross-recurrence plots in the rightmost column. Building on the first set of examples, we included two Van der Pol systems’ with two different damping parameters. The non-dampened system (blue, solid line) provides a comparison to the examples provided in Figure 1. The second and third sets of examples (second row) utilize the same damping coefficients, however, one of the systems shown in the third set of examples (third row) inverts one of the the systems to provide a visual example of how such a modification will impact the state space reconstruction and cross-recurrence plot. These modifications driven by the damping coefficients are included to illustrate how changes to one’s internal dynamics (e.g., nerve conduction rate, motor plan selection) may alter the dynamic patterns exhibited at the behavioral (movement) level, and thus, the shared dynamics between the two observed systems.

### 5.2. Empirical Data: EEG and Center of Mass Acceleration during Walking

To provide an example of the processing methods associated with a dynamical systems approach to quantifying BBIs we include data from two sample subjects (H05, H07). These data are provided as a means to demonstrate real-world BBIs measured using EEG and IMUs and modeled according to the above theoretical framework, but without any intention to derive statistically supported or clinically relevant conclusions. All protocols were approved by the University of Utah Institutional Review Board.

### 5.3. Data Acquisition

EEG data were collected from a 32-channel ActiCap with sintered Ag/AgCl sensors and a wireless LiveAmp (Brain Products, Gliching, Germany) amplifier. Baseline impedances were ≤25 kOhm and data were recorded at 250 Hz (BrainVision Recorder, Brain Products) during 170 s of walking at a preferred pace back and forth between two lines (7 m apart) in a well-lit biomechanics laboratory (12 m × 7 m). Approximated center of mass (COM) motion was collected via inertial measurement units (IMU) (Opal, v2, APDM Inc., Portland, OR, USA) that were secured over the lumbar spine with an elastic waist belt and recorded a sampling rate of 128 Hz. Data were synchronized at the beginning of each recording using a TTL pulse delivered from the IMUs, using Moveo Explorer (APDM Inc.), to the LiveAmp via a 2.5 mm jack trigger port. This synchronization required tethering the LiveAmp to the external sync box of the IMUs; but the systems were untethered (i.e., wireless) after the sync was delivered and before walking began.

#### 5.3.1. EEG Data Processing

EEG data were processed offline in Matlab (2022a, Mathworks, Natick, MD, USA) with tools implemented in EEGLAB [110] and Fieldtrip [111] toolboxes. Briefly, data were imported (*pop_loadbv*) and filtered (*pop_eegfiltnew*) to 1–50 Hz. Adaptive mixture independent component analysis (AMICA) was used (*runamica15*) in conjunction with an automatic classifier (*ICLabel*) to remove components that were classified as non-brain (i.e., ocular, EMG, EKG, channel noise, line noise, or ‘other’). Noisy channels were removed if they were poorly correlated with their estimate, as reconstructed from neighboring channels. Because this step was performed after an initial ICA step (thereby removing similar parts of the signal across channels), and because dynamic movement is known to induce inhomogenous noise across scalp electrodes, we selected a threshold of r ≤ 0.6 for this step, which is more liberal than the default threshold of r ≤ 0.85. Removed channels were subsequently interpolated using a spherical spline and the data were subjected to a second AMICA to again remove all non-brain components, after rank adjustment for the interpolated channels.

EEG data were re-referenced to the average scalp potential to account for the effects of volume conduction. To match the resolution of COM data, continuous EEG data were upsampled to 256 Hz (ft_resampledata) and epoched by creating a sliding window of 512 samples (2 s) long with 99.6% overlap to achieve a functional resolution of 128 Hz. Time-varying spectral power density was quantified (ft_freqanalysis) for each epoch using a multi-taper method based on discrete prolate Slepian sequences [112] from 1–50 Hz with a smoothing factor of 3 Hz. Time series data were extracted from six channels approximating the left somatomotor cortex (FC5, C3, CP5) and right somatomotor cortex (FC6, C4, CP6) to represent oscillatory activity in alpha (8–13.5 Hz), beta (14–29.5 Hz), and low gamma (30–50 Hz) bands. The power spectra across these steps is depicted in Figure 3. The power spectra (mean +/−95%CI) for each subject across all channels and windows is displayed in Figure 4.

#### 5.3.2. COM Data Processing

Approximate center of mass data were obtained in sensor-fixed coordinate systems and rotated into a body-fixed reference frame that was initially aligned with the global reference frame. The resulting kinematics yielded body-fixed linear accelerations in the anteroposterior (AP), mediolateral (ML), and vertical (VT) directions. Accelerations were filtered using a fourth-order Butterworth filter with an 8 Hz cutoff frequency. A resultant was calculated using AP, ML, and VT to examine the coupling between total acceleration and the EEG readings. All analyses related to the dynamics of these systems and the coupling between them were performed in Python [113] (v3.10.9). Specific modules include NumPy [114], pandas [115], SciPy [116], PyRqa [117], and EntropyHub [118]. Time series data from EEG and COM are provided in Figure 5. In addition to the channel-wise data, averages for the left and right hemispheres were calculated for each band (i.e., SCO, alpha, beta, and gamma) (Figure 4, first and second columns). An example of the center of mass data from the anteroposterior (AP), mediolateral (ML), vertical (VL), and resultant (RES) accelerations are provided in Figure 4 (third column).

Descriptive plots of EEG data (Figure 3 and Figure 4) are rarely reported in BBI literature, with most authors depicting only statistical results (e.g., changes in time-locked spectral perturbations across group-averaged gait cycles). Given the lack of standards in processing EEG data, such reporting could improve transparency about (i) initial signal quality and (ii) the effect (if any) of preprocessing on the power spectra. It is clear from Figure 3 that the pre-processing steps (filtering and blind source separation (ICA)) had the greatest effect on slow-cortical oscillations (SCOs), which is to be expected given the overlap with the frequency of a typical gate cycle. Although none of the channels-of-interest were identified as being of poor quality, we justified the selection of a more liberal threshold for bad-channel detection than is typically employed. Thus, the fast gamma ‘spiking’ observed in all channels (Figure 4) may be due to movement and thus, may be an indicator of poor signal quality. One solution proposed by Nordin and colleagues (2018) is to collect data concomitantly from externally facing ‘noise’ electrodes. These signals may be particularly effective at cleaning the types of artifacts we observed here, yet this equipment is not universally available. Moreover, automatic subspace reconstruction (ASR), a popular artifact removal technique, was not applied to these data since acquisition in the current study only included 3 s of standing ‘rest’, which is generally considered to be insufficient to serve as ‘calibration’ data (e.g., ~1 min of data acquired at rest).

#### 5.3.3. Parameter Selection

Proper determination of the time delay, *τ,* and embedding dimension, *m*, are essential components of assessing the dynamics of a time series. For thoroughness, we provide the autocorrelation and average mutual information of the time series in Figure 5 (additional examples are provided in SM3 and SM4). Considering the data from both subjects, we have chosen a time delay of *τ* = 16 to coincide with a plateau in the AMI plots and the interpretability of this value relative to the 128 Hz sampling rate. An embedding, *m* = 3, was chosen for the state space reconstruction based on the limited reduction in the percentage of false nearest neighbors at higher embeddings (Figure 6). The state space reconstruction of EEG and COM data for each subject is provided in Figure 7, having used *m* = 3 and *τ* = 16.

The distinct patterns observed in COM data are expected from this type of task. The attractor pattern in AP, ML, and VT accelerations can be attributed to more or less consistent movements in those planes from step-to-step, with deviations attributed to the natural perturbations in the gait cycle. Patterns in the EEG data are less intuitive. There are clear perturbations in the system that appear to differ by band. On one hand, the patterns are similar between hemispheres, which would be expected during a complex, bipedal task such as overground walking. On the other hand, the systems appear to exhibit a random state-space trajectories. This is consistent with early efforts revealing that EEG rhythm dynamics cannot be distinguished from filtered noise (modeled using surrogate/shuffled data) [119]. However, despite their random appearance, state-space reconstructions can distinguish seizure EEGs from resting state EEGs [120] and are believed to represent system multistability, a property defined by having multiple stable states (attractors). This would be expected from brain motor circuits that must constantly adapt to maintain coordination across multiple systems to maintain a behavior as complex as overground walking [121]. Taken together, the randomness embedded in these reconstructions supports that scalp-measured EEG activity represents complex interactions between many interacting generators (rather than a single generator), while the dynamic and deterministic aspects may allow for valuable insights into the nature of those interactions important for neuromotor control.

Although deviations from attractor patterns (i.e., large distances between cycles) may be meaningful, it is also noteworthy that the most prominent deviations are seen in reconstructions of SCOs (subject H05) and reconstructions of gamma band activity (both subjects). It was previously noted in this review that activity in these bands may suffer the most from movement-related artifacts. Therefore, it may be possible to interpret these plots as containing valuable information about signal quality. We recommend future work to explore the use of state space reconstructions as part of an iterative denoising pipeline to remove parts of the signal that lead to implausible state-space trajectories.

#### 5.3.4. Sample Entropy and Recurrence Quantification Analysis

The regularity and dynamics of the univariate time series were examined with sample entropy and recurrence quantification analysis. Sample entropy was calculated on the EEG and COM data across a range of embedding dimensions, *m* = [{2, 3, 4, 5}, and radii, *r* = {0.1, 0.15, … 0.4} with a fixed time lag, *τ* = 16 (Figure 8 and Figure 9). For a given template length, sample entropy typically falls as the tolerance, *r*, is increased. This behavior can be observed in both subjects when *m* = {2, 3, 4} and *r* > 0.15. However, we highlight the rise in sample entropy for *r* = {0.1, 0.15} for H07 (e.g., SCO, alpha, and beta). This response is likely a function of walking frequency and the combined choice of *τ* = 16 and *m* = {4, 5}. A similar, more exacerbated, response can be observed in the COM data (Figure 9) where templates *m* > 2 follow the same behavior. Comparisons for EEG and COM data across both subjects are provided in Figure 10. In addition to the sample entropy measures for the EEG and COM data, we provide the percent recurrence from RQA to further accentuate how the behaviors of these systems can be observed through a variety of techniques. Sample entropy and recurrence quantification analysis are different techniques that necessitate different steps and considerations; however, the concurrent use of these tools can provide several benefits to overall interpretation—especially in subsequent steps where the shared dynamics of these systems are considered.

Based on the patterns observed in Figure 8 and Figure 9, specifically the instability of the sample entropy measure at higher template lengths and lower radii, templates of *m* = 3 and *m* = 2 were chosen to quantify the regularity of EEG and COM data, respectively (with *r* = 0.2 and *τ* = 16 in all cases). To aid with visualization, we scale sample entropy to the peak sample entropy value at the subject level, providing a relative reference to peak irregularities between EEG and COM data for each subject.

Because different template lengths were selected to assess signal regularity, and sample entropy values were scaled for each subject and each system (EEG, COM), we cannot directly compare sample entropy between subjects. However, the relative patterns of complexity and dynamics of the COM data (Figure 10) appear similar: The ML accelerations exhibited the greatest complexity while AP accelerations exhibit the greatest stability. As expected, this indicates a more repetitive structure of movement in the AP (and VT) directions and less structured movement in the ML direction. The high degree of recurrence in VT accelerations makes intuitive sense given that the vertical trajectory is highly stable in overground walking. Likewise, both subjects exhibited greater complexity in slower rhythms (SCO and alpha bands) and greater stability in gamma rhythms.

The patterns for SampEn and REC are largely complementary in that SCOs exhibited the lowest REC/greatest SampEn and gamma band activity exhibited the greatest REC/lowest SampEn in both participants. Beyond the possibility that this pattern indicates movement-based artifact, this pattern is consistent with prior reports of coupling between high-gamma activity across an error-detection network (i.e., anterior cingulate, posterior parietal cortex, and sensorimotor cortex) and the gait cycle [49]. Considering the biophysical interpretations discussed in the first part of this review, these patterns are suggestive of flexible long-range interactions between the motor cortex and the rest of the brain that constrains locally coordinated activity.

#### 5.3.5. Coupling between Brain and Body Systems

With a better understanding of the dynamics and regularity of the univariate EEG and COM data, we shift our focus to the coupling between these two systems with cross-sample entropy and cross-recurrence quantification analysis. As was performed in the univariate case, cross-sample entropy for left and right hemispheres were calculated across a range of template lengths, *m* = {2, 3, 4, 5}, and radii, *r* = {0.1, 0.15, …, 0.4}, and *τ* = *16* (Figure 11). Notably, the behavior of the cross-sample entropy statistic follows more closely to the expected behavior when changing *m* and *r*. Based on these data, a comparison of cross sample entropy between subjects (*m* = 3, *r* = 0.2, *τ* = 16) is provided in Figure 12.

The cross-recurrence plots for combinations of EEG (i.e., alpha, beta, gamma, SCO) and COM (i.e., AP, ML, VT, RES) data from a single subject are provided in Figure 13 (two additional subjects provided in SM7 and SM8). Unlike the cross-sample entropy approach, cross-recurrence plots can be analyzed by a bevy of different, complementary metrics. Determinism (DET)—the percent of the plot comprised of diagonal lines—and recurrence rate (REC)—the percent of the plot that contains an intersection of the systems—are the most common and intuitive values reported in the cross-recurrence literature. In other words, REC and DET represent the number of recurrences and the duration of those recurrences, both interpreted as stability in coupling between two systems. We provide a visual comparison between hemispheres and across subjects for both the percent determinism and recurrence in Figure 14.

Visual inspection of the cross-sample entropy plots (Figure 12) reveals clear differences in the coupling between EEG and COM accelerations across participants that are unique to each participant, unlike the more similar between-subject patterns revealed from the EEG-only and COM-only plots discussed to this point. Perhaps most notable is the lesser regularity (greater cross-SampEn) and shorter periods of regularity (lesser DET) between broadband EEG complexity and ML acceleration complexity. The lack of distinct EEG band-specific patterns of synchrony may represent an artifact of combining signals across different channels (three in each hemisphere), whereas explicit modeling of interactions between channel-wise signals may reveal interesting patterns [122]. Of course, following the biophysical origins reviewed earlier, it is reasonable to expect that information flows across all microcircuits—across laminar divisions, in both a top-down and a bottom-up fashion, and between thalamus and cortex—to support and regulate a behavior as complex as overground walking. On the other hand, this pattern may represent the functional relevance of broadband power or the ‘offset’ in these combined channels during volitional movement. There is a growing interest in two so-called ‘aperiodic’ properties of the EEG spectrogram, the offset and the aperiodic exponent, as the rate of decay that follows the 1/f scaling of the power spectra [21]. It is not uncommon to correct for broadband activity by scaling spectral power by the offset, yet this offset appears to be directly related to cortical activity [123,124]. Future studies employing these methods on larger samples may consider exploring this possibility by quantifying coupling between dynamic aperiodic activity and kinematics.

Notwithstanding, a subtle deviation from these broadband patterns is observed between the complexity of VT accelerations and greater stability (REC, cross-SampEn) and longer duration of stability (DET) in coupling with faster rhythms (beta, gamma) than with slower rhythms (SCOs, alpha). It is possible that this represents a step-by-step braking mechanism, such that the motor cortex is being primed to adapt to new sensory information with each step. Given that this mechanism appears to deteriorate with age [125], comparing these patterns between young and old groups is warranted.

## 6. Future Directions

There is increasing interest to study the brain during dynamic movement and to quantify BBIs, yet most of this work has defined those interactions through statistical, linear dependencies between the constituent signals themselves. These approaches have revealed interesting and replicable patterns, yet the biological basis for those patterns and their interpretation in the context of a dynamical systems approach to human movement is unclear. In this review, we sought to (i) synthesize what is known about the biophysical interpretation of scalp-measured rhythms from EEG, (ii) advise on best practices for ‘cleaning’ those data, and (iii) introduce the reader to the proper use of two tools—cross-sample entropy and recurrence quantification analysis—that may be useful for future BBI studies. To the authors’ knowledge, the worked example included with this review represents a novel approach to modeling BBIs through a dynamical systems lens. Through this example, we revealed patterns of coupling that were common across two participants, but the extent to which they represent healthy or disordered neuromotor control remains an open topic worthy of further exploration. Moreover, although this review focused on BBIs along a relatively short time scale in a laboratory environment, biological signals operate across a variety of scales and frequencies, including infradian rhythms lasting longer than 24 h. Thus, BBIs in these longer time scales measured in unconstrained, free-living environments warrant further investigation.

## Figures and Tables

**Figure 1 sensors-23-06296-f001:**
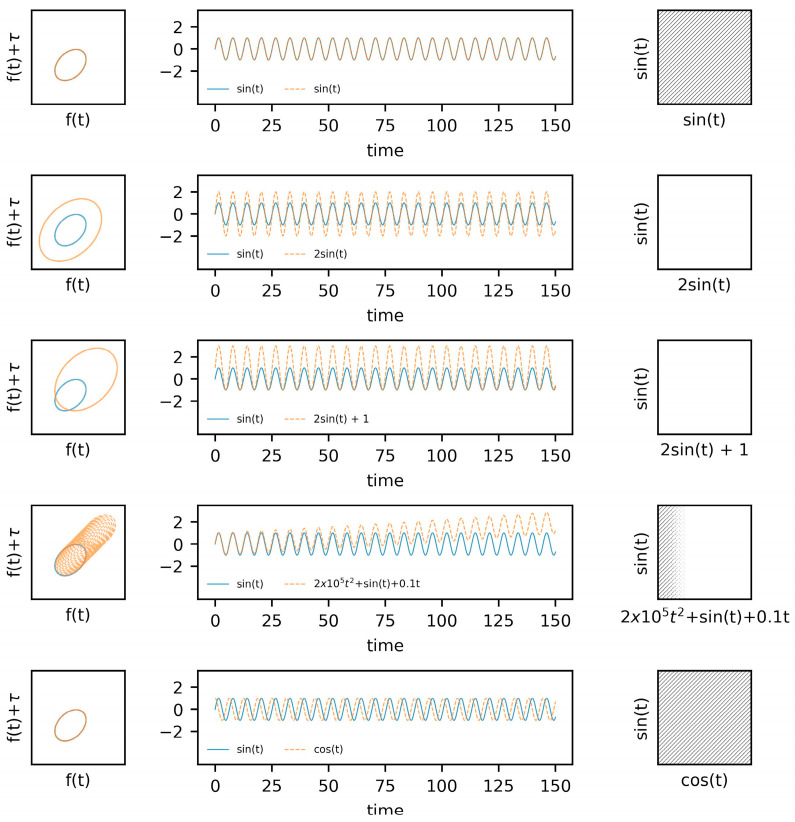
Examples of simple limit cycles overlaid in state space, time, and in cross recurrence.

**Figure 2 sensors-23-06296-f002:**
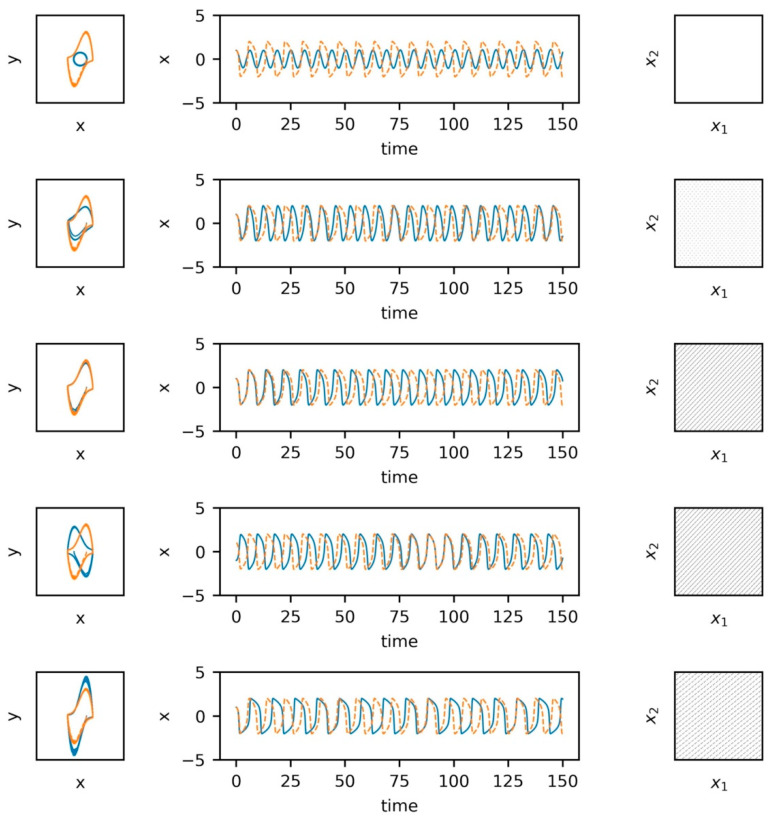
Examples of Van der Pol system overlaid in state space, time, and in cross recurrence.

**Figure 3 sensors-23-06296-f003:**
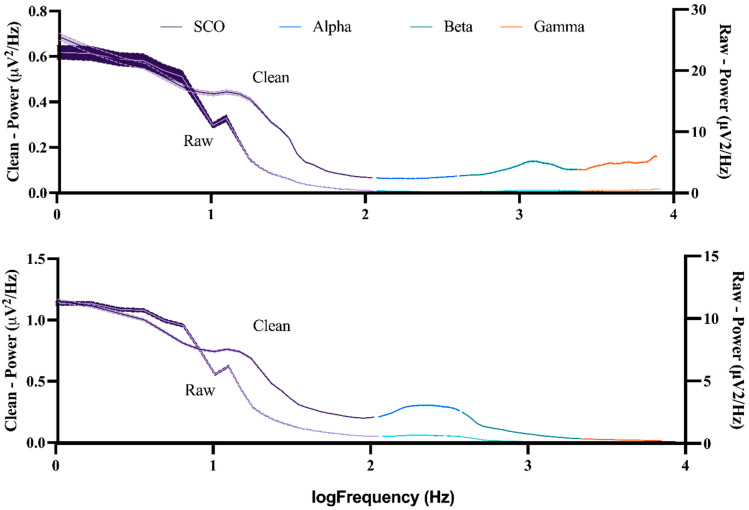
Mean spectral power density for slow cortical oscillations (SCO), alpha, beta, and gamma rhythms (shaded bars are 95% CI) for H05 (top) and H07 (bottom). Raw signals (right axis) and clean signals (left axis) are averaged over all channels-of-interest (k = 6) and all windows (~21,300; each 2000 ms long).

**Figure 4 sensors-23-06296-f004:**
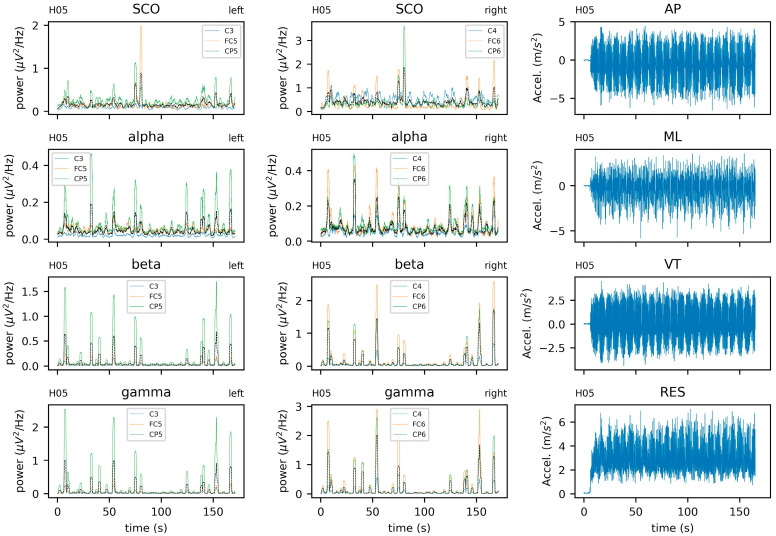
Electroencephalogram (EEG) and center of mass (COM) data from a single subject (H05) EEG data include alpha, beta, gamma, and combined theta/delta (SCO) bands. Black lines represent averages across three intrahemispheric channels. COM data include anteroposterior (AP), mediolateral (ML), vertical (VT), and resultant (RES) accelerations. Data from the additional subject (H07) is provided in the Appendix A.

**Figure 5 sensors-23-06296-f005:**
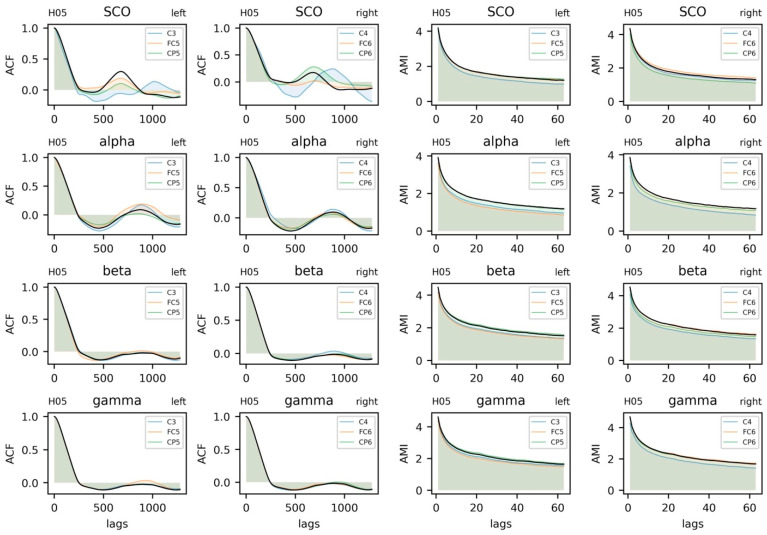
Autocorrelation and average mutual information functions for electroencephalogram (EEG) data from a single subject. EEG data include alpha, beta, gamma, and combined theta/delta (SCO) bands. Black lines represent averages across three intrahemispheric channels.

**Figure 6 sensors-23-06296-f006:**
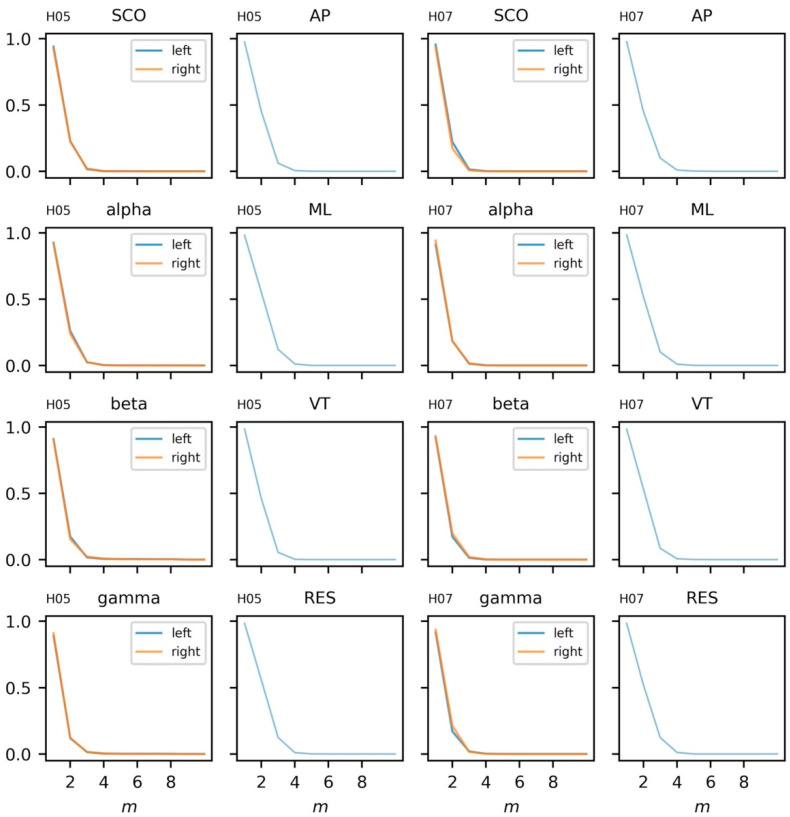
Percent false nearest neighbors estimations for electroencephalogram (EEG) and center of mass (COM) data from each subject (columns). EEG data include alpha, beta, gamma, and combined theta/delta (SCO) bands averaged across three intrahemispheric channels. COM data include anteroposterior (AP), mediolateral (ML), vertical (VT), and resultant (RES) accelerations.

**Figure 7 sensors-23-06296-f007:**
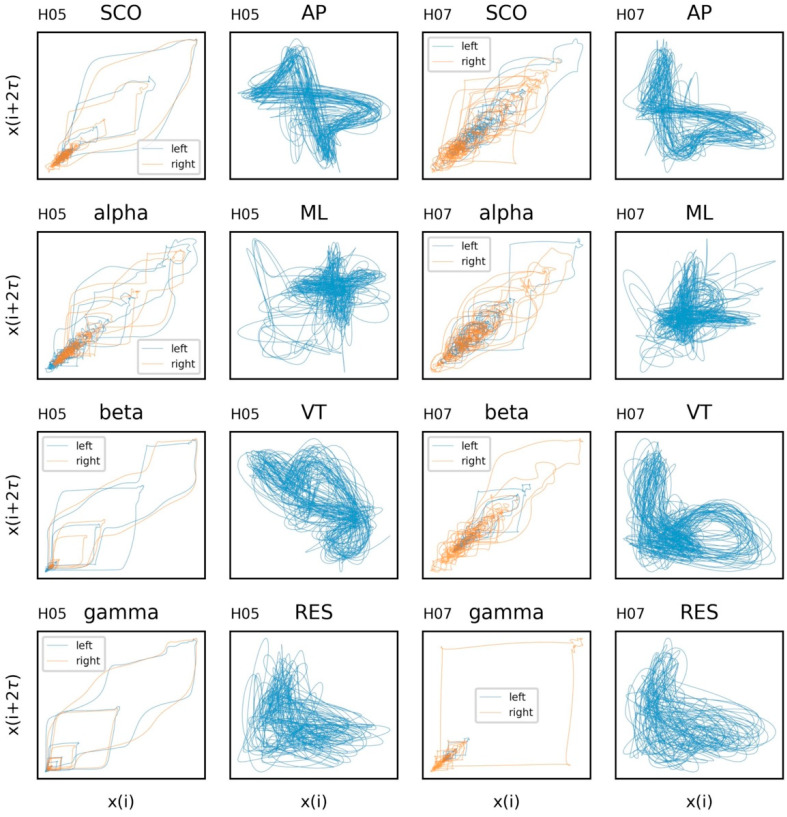
State space reconstruction for electroencephalogram (EEG) and center of mass (COM) data from each subject (columns). EEG data include alpha, beta, gamma, and combined theta/delta (SCO) bands. COM data include anteroposterior (AP), mediolateral (ML), vertical (VT), and resultant (RES) accelerations. {*m* = 3, *τ* = 16}.

**Figure 8 sensors-23-06296-f008:**
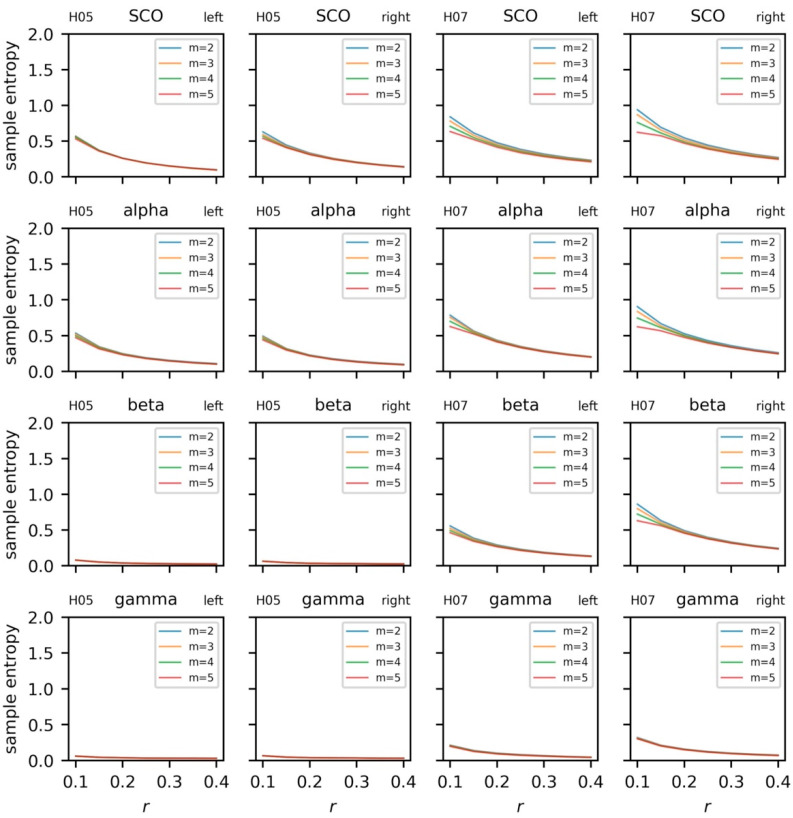
Sample entropy for electroencephalogram (EEG) data from each subject (columns). EEG data include alpha, beta, gamma, and combined theta/delta (SCO) bands averaged across three intrahemispheric channels.

**Figure 9 sensors-23-06296-f009:**
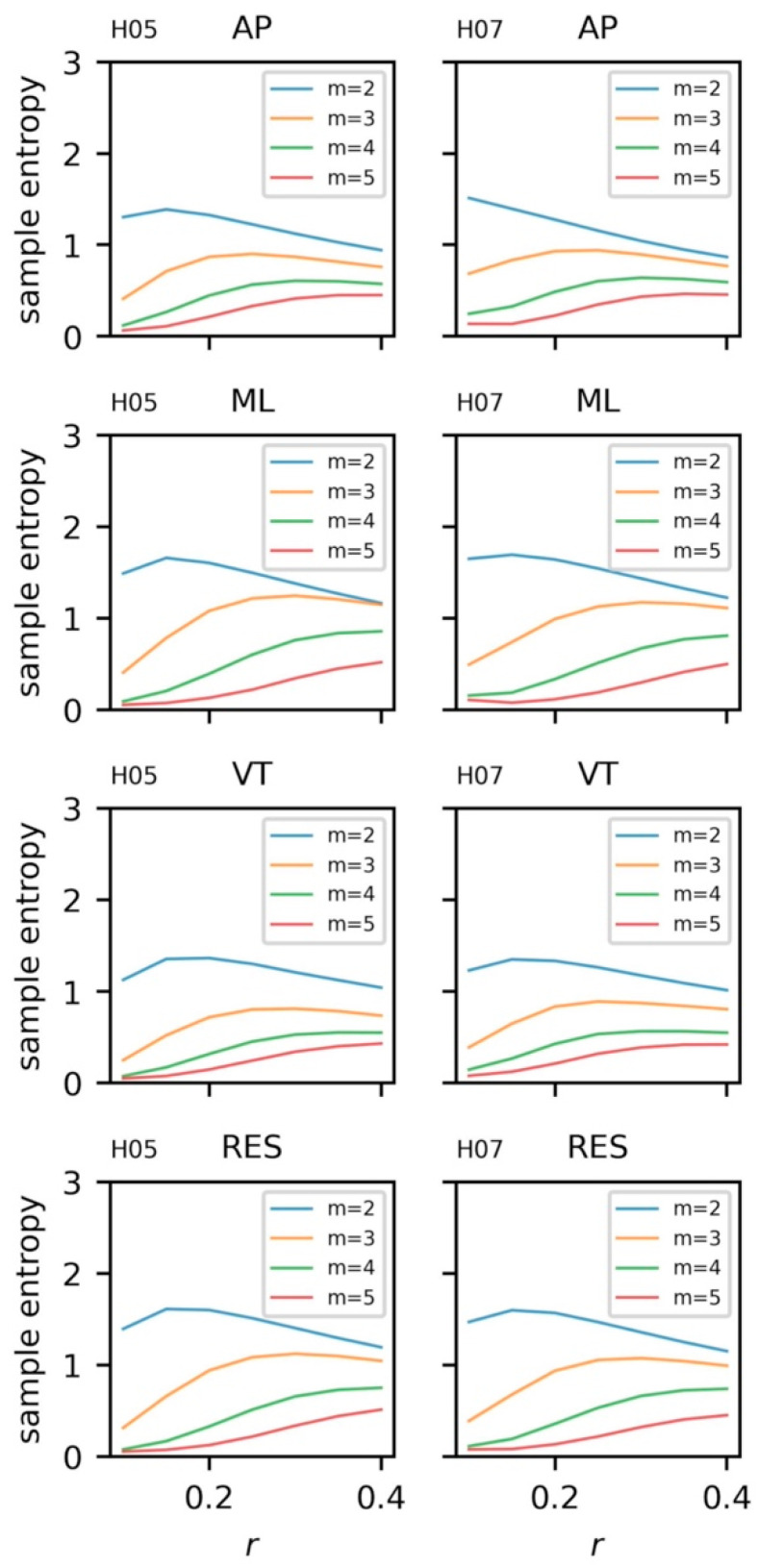
Sample entropy for center of mass (COM) data from each subject (columns). COM data include anteroposterior (AP), mediolateral (ML), vertical (VT), and resultant (RES) accelerations. {*m* = (2, 3, … 5), *r* = (0.1, 0.15, … 0.40), τ = 16}.

**Figure 10 sensors-23-06296-f010:**
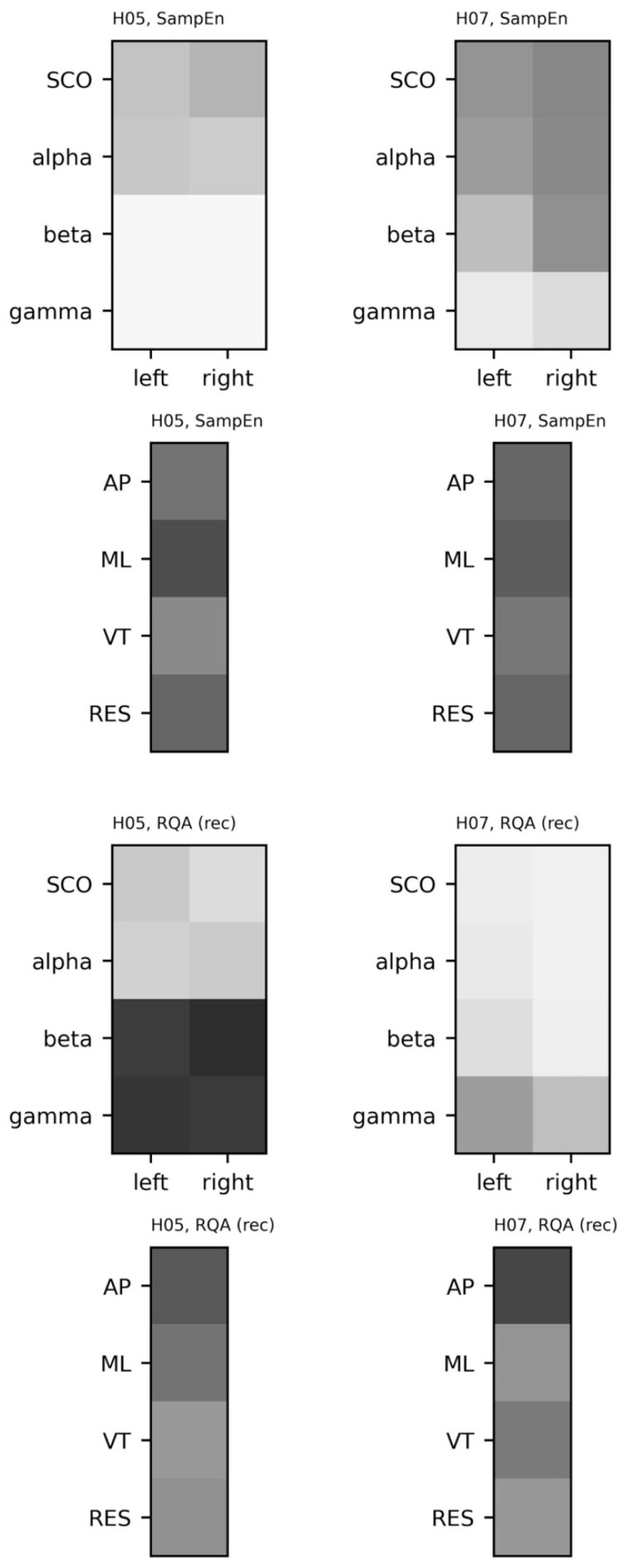
Sample entropy and percent recurrence (RQA) for electroencephalogram (EEG) data from each subject (columns). EEG data include alpha, beta, gamma, and combined theta/delta (SCO) bands. COM data include anteroposterior (AP), mediolateral (ML), vertical (VT), and resultant (RES) accelerations. Sample Entropy: {EEG: *m* = 3, *r* = 0.2, *τ* = 16; COM: *m* = 2, *r* = 0.2, *τ* = 16}. RQA: {EEG: *m* = 3, *r* = 0.3, *τ* = 16; COM: *m* = 3, *r* = 0.75, *τ* = 16}. Data were scaled relative to the highest value observed across subjects; darker shades represent higher sample entropy and recurrence rates from RQA.

**Figure 11 sensors-23-06296-f011:**
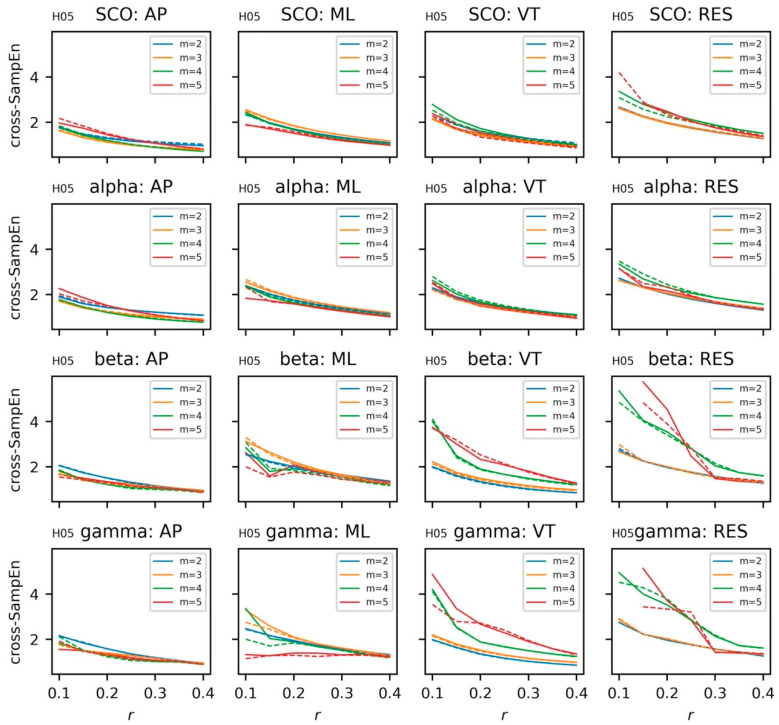
Cross-sample entropy for electroencephalogram (EEG) data from a single subject. EEG data include alpha, beta, gamma, and combined theta/delta (SCO) bands. COM data include anteroposterior (AP), mediolateral (ML), vertical (VT), and resultant (RES) accelerations. Solid lines denote the left hemisphere; dashed lines denote the right hemisphere. {*m* = (2, 3, … 5), *r* = (0.1, 0., … 0.40), τ = 16}. Plots for the additional subject (H05) can be found in the Appendix A.

**Figure 12 sensors-23-06296-f012:**
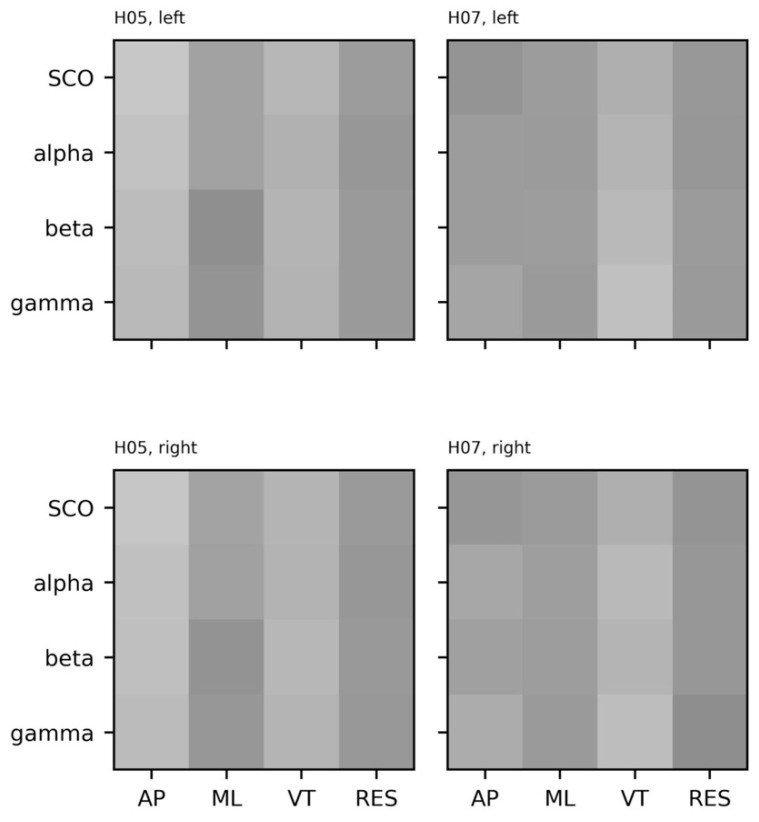
Heatmaps for the cross-sample entropy for electroencephalogram (EEG) and center of mass (COM) data from each subject (columns). EEG data include alpha, beta, gamma, and combined theta/delta (SCO) bands. COM data include anteroposterior (AP), mediolateral (ML), vertical (VT), and resultant (RES) accelerations. {*m* = 3, *r* = 0.2, τ = 16}. Data were scaled relative to the highest value observed across subjects; darker shades represent a higher cross-sample entropy (i.e., more asynchronous patterns between EEG and COM data).

**Figure 13 sensors-23-06296-f013:**
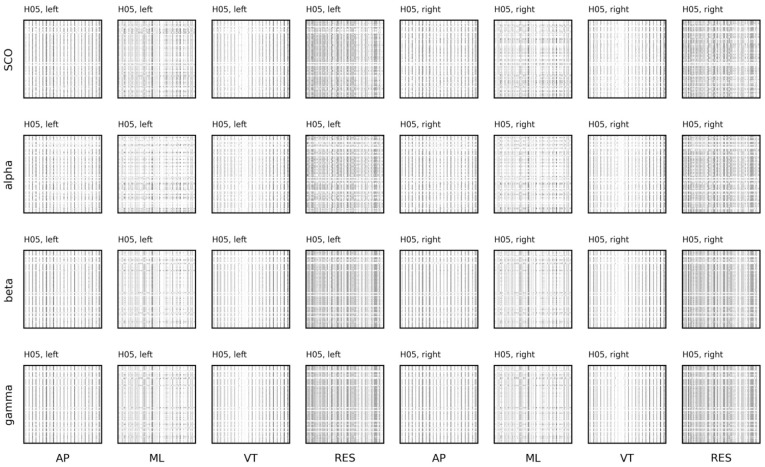
Cross-recurrence quantification (cRQA) for electroencephalogram (EEG) and center of mass (COM) data from a single subject. EEG data include alpha, beta, gamma, and combined theta/delta (SCO) bands. COM data include anteroposterior (AP), mediolateral (ML), vertical (VT), and resultant (RES) accelerations. {*m* = 3, *r* = 0.2, τ = 16, *tw* = 1}. Plots for the additional subject (H05) can be found in the Appendix A.

**Figure 14 sensors-23-06296-f014:**
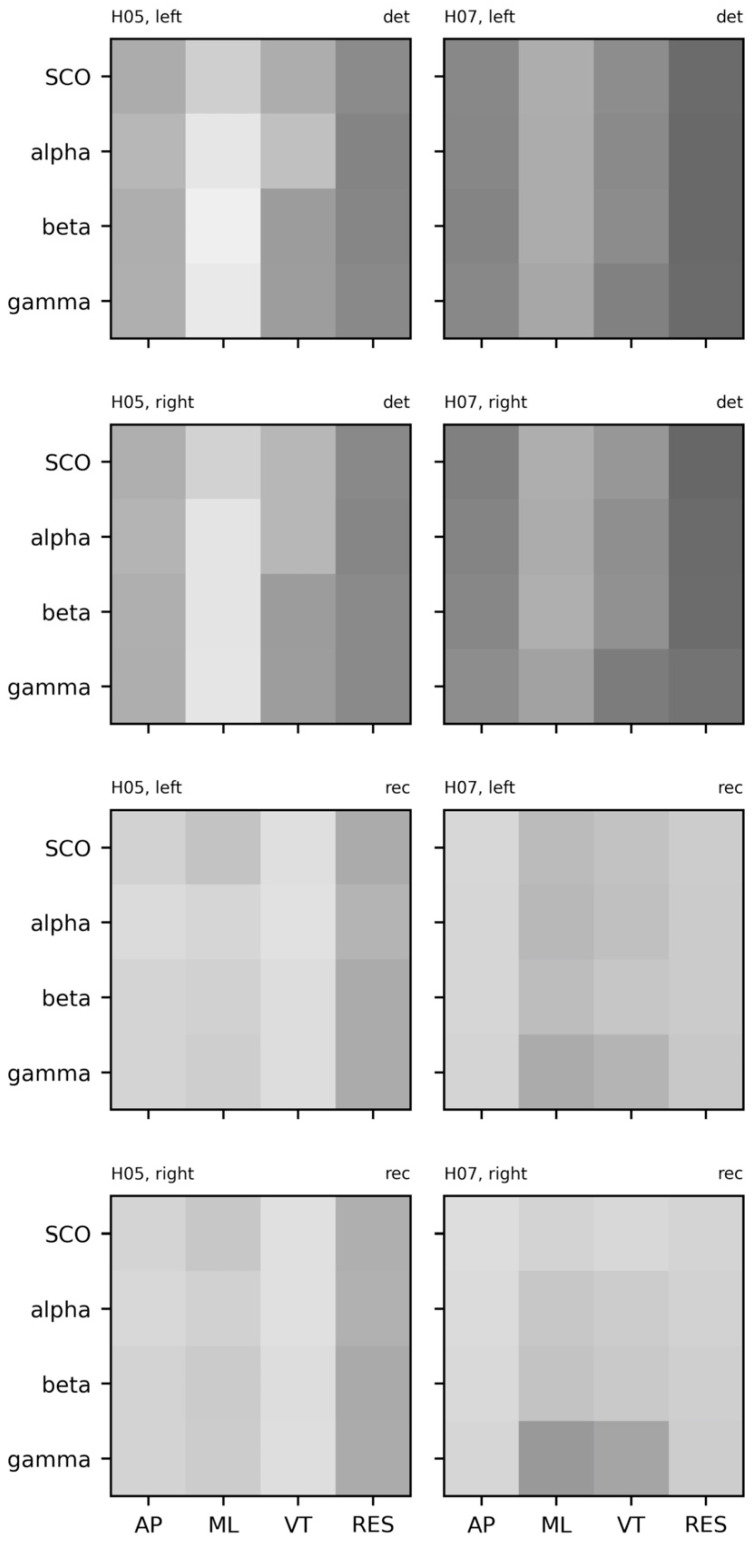
Heatmaps of percent determinism (det) and percent recurrence (rec) from cross-recurrence quantification for electroencephalogram (EEG) and center of mass (COM) data from each subject (rows). EEG data include alpha, beta, gamma, and combined theta/delta (SCO) bands averaged across three intrahemispheric channels. COM data include anteroposterior (AP), mediolateral (ML), vertical (VT), and resultant (RES) accelerations. {*m* = 3, *r* = 0.2, τ = 16, *tw* = 1}. Data were scaled relative to the highest value observed across these subjects; darker shades represent higher determinism and recurrence, respectively.

## Data Availability

The data and code used to generate the results discussed herein are openly available at https://github.com/uncg-kines/bbi-review-2023 (Accessed on 1 May 2023).

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
