# Peer review of "A Dynamical Systems Approach to Characterizing Brain–Body Interactions during Movement: Challenges, Interpretations, and Recommendations"

_sensors, 2023, doi:10.3390/s23146296_

Round 1
Reviewer 1 Report
In this study, the authors analyzed brain-body interactions from the viewpoint of a dynamical systems. The overall aim and outline of the manuscript are clear. I have some comments as follows.
1. The EEG sample seems a bit small. Has an a priori power analysis been conducted? Is analysis reliable with this small sample?
2. In Line 480, please check grammar for the sentence “we include data three sample subjects”.
Author Response
In this study, the authors analyzed brain-body interactions from the viewpoint of a dynamical systems. The overall aim and outline of the manuscript are clear. I have some comments as follows.
1. The EEG sample seems a bit small. Has an a priori power analysis been conducted? Is analysis reliable with this small sample?
We appreciate the opportunity to clarify an important point. The third part of this review is intended to be an example of the types of analyses and metrics that can be derived when examining coupling between EEG (‘brain’) and IMU (‘body’) signals through a dynamical systems lens. Given the paucity of published analyses using similar approaches, we believe the examples and accompanying code will be of great interest to a broad audience. In the original manuscript we stated the following:
“These data are provided as a means to demonstrate what real-world BBIs measured using EEG and IMUs and modeled according to the above theoretical framework, look like, but without any intention to derive statistically supported or clinically relevant conclusions.”
In conjunction with our response to concerns raised by the Associate Editor, we have reduced the examples down to just two subjects, thus removing any speculative language regarding the nature of TBI or concerns regarding statistical power. Thus, the paper is now better aligned with our intention of providing data as a worked example of how two related but distinct analytical approaches (that are novel in this field) could address BBIs, rather than using the data to make inferences on what the interactions between EEG and IMU data mean for health, disease, or injury. Our aim was to write this review as a roadmap for others who could use this approach in their appropriately powered datasets to make such inferences. We hope that the reviewer finds this third section to be more aligned with our ultimate goal.
2. In Line 480, please check grammar for the sentence “we include data three sample subjects”.
Thank you for bringing this error to our attention. We have checked the entire manuscript, including this line, for grammatical errors. That line now reads:
“To provide an example of the processing methods associated with a dynamical systems approach to quantifying BBIs we include data from two sample subjects (H05, H07).”
Reviewer 2 Report
This is an exciting and thorough study; it is a good attempt to understand the biological basis and interpretation of replicable patterns in the context of human movement and brain signals. The manuscript is well written.
Author Response
We thank the reviewer for their careful and thoughtful consideration of the manuscript.
Reviewer 3 Report
Review Report
Journal Sensors (https://www.mdpi.com/journal/sensors) (ISSN1424-8220)
Manuscript ID sensors-2407885
A Dynamical Systems Approach to Characterizing Brain-Body 2 Interactions During Movement: Challenges, Interpretations, 3 and Recommendations
In General ..
Well written review-type survey article.
Very comprehensive,
and long list of references.
No clear figures this is to be fixed.
There some repeated and well known facts from the EEG community point of views, this is related to:
2. EEG Signal Acquisition and Preprocessing. All these terminologies are well known.
2.1. Re-Referencing 66, 2.2. Artifact Removal, 3. Interpretation of EEG-Measured Oscillatory Activity 142, …
I do not know why the authors are repeating this in this article. It could be shorted ..
For the (To the authors’ knowledge, the worked example included with this review represents a novel approach to modeling BBIs through a dynamical systems lens)
Any reason the authors did the study in terms of the dynamic system point view. This must be well clarified.
Author Response
In General ..
Well written review-type survey article.
Very comprehensive,
and long list of references.
1. No clear figures this is to be fixed.
Thank you for bringing this to our attention. We have ensured that high-resolution figures now accompany our manuscript.
2. There are some repeated and well known facts from the EEG community point-of-view, this is related to: EEG Signal Acquisition and Preprocessing (all these terminologies are well known), 2.1. Re-Referencing, 2.2. Artifact Removal, 3. Interpretation of EEG-Measured Oscillatory Activity. I do not know why the authors are repeating this in this article. It could be shortened.
We are grateful to have an expert in the EEG field reviewing our manuscript, and we agree that among those doing this work (and in the broader EEG community) these concepts and terminologies are well-known. In constructing this review of the literature for a multidisciplinary journal (‘Sensors’) and in particular for a Special Issue that is not specific toward any imaging modality (‘Sensors in Neuroimaging and Neurorehabilitation’), we feel it is important to write this for a broad audience. For example, an expert who measures movement kinematics and biomechanics (e.g., with an IMU) should be able to understand and appreciate the intricacies of processing and interpreting EEG data. We now motivate this in the introduction to this section as follows:
“New analytic tools for this task are published often, which can be overwhelming to novice investigators and non-experts. In the following sections, we outline two of the most frequently employed pre-processing steps (re-referencing and artifact removal) and highlight existing recommendations for their use in studies that use scalp-measured EEG to study BBI.”
Regarding section 2.3: to our knowledge there is no similar, recent review of EEG biophysics through this lens. It is important that these details are explained so as to move the field away from interpretations of scalp-measured EEG rhythms based on behavior alone (e.g., “Alpha power represents focus, relaxation”), a practice which is rampant, particularly in the movement sciences. We have expanded this section to address somewhat opposing concerns (of lacking, rather than redundant information) raised by the Associate Editor. We now motivate this section as follows:
“For the most part, in studies of BBI using scalp-measured EEG, a priori feature selection is based on prior findings from the BBI literature (e.g., a focus on slow-rhythm oscillations during walking without consideration for faster rhythms) or on the response of those features to cognitive demands (e.g., alpha rhythms during an attentional task). In the following sections we aim to distill data from biophysical models that elucidate more precisely the circuit interactions that give rise to these rhythms. It is worth noting that few of the studies cited herein involve dynamic, complex movements and mostly rely on distal upper limb (finger, wrist) movements. We posit that the neuroanatomical control of these finer-grain movements at the scale accessible to scalp-measured EEG is not different enough to warrant their lack of inclusion of BBI studies based on complex movements. To the contrary, we believe this underscores the importance of this review and motivates a pivot toward interpreting BBIs during complex movements based on these models (and not strictly on behavioral corelates).”
3. For the statement “To the authors’ knowledge, the worked example included with this review represents a novel approach to modeling BBIs through a dynamical systems lens(...)”, is there any reason the authors did the study in terms of the dynamic system point view? This must be well clarified.
We appreciate the opportunity to expand our rationale given that this is critical to motivating the entire review. In the original manuscript, we introduce a mathematical rationale between Lines 286-300. We expand with additional intuition:
“The differences between these approaches are not merely mathematical: they represent divergent ways of thinking about motor behavior. For example, in the classical sense, movement is believed to arise from a pattern generator or collection of controllers. This leads to an interpretation of movement variability (e.g., stride time variability during over ground walking, reaction time variability in response to a cognitive task) as representing error arising from maladaptive organization of the underlying cognitive or anatomical structures (controllers). On the other hand, a dynamical systems approach leads to an interpretation that some degree of variability is ‘normal’, representing adaptive system organization(Riley & Turvey, 2002; Stergiou & Decker, 2011) . The differences become even more nontrivial when considering how to relate measures of each system (e.g., EEG-measured oscillatory activity and accelerometer-measured movement kinematics). Movement patterns are not governed strictly by output from the nervous system, but by information, which is in turn generated from a recurrent network engaging sensory and motor anatomical networks and cognitive processes. The interactions between systems are inherently nonlinear and are characterized by metastability, representing flexibility of the system around an attractor state, which cannot be captured using the traditional linear (‘effect=structure’) approach (Van Orden et al., 1997). This acknowledgement led to growth in the fractal physiology field (Goldberger & West, 1987; West, 2021), which adopts a dynamical systems approach to examine changes within and between systems. Therefore, the focus and foundation of thought presented herein is grounded in dynamical systems theory.”